Genome-wide analysis of the lignin toolbox for morus and the roles of lignin related genes in response to zinc stress

Chao Nan 1 2
Yu Ting 1
Hou Chong 1
Liu Li 1 2
Zhang Lin 1 2 moruszhanglin@126.com
1 Jiangsu Key Laboratory of Sericultural Biology and Biotechnology, School of Biotechnology, Jiangsu University of Science & Technology , Zhenjiang, Jiangsu Province , China
2 Sericultural Research Institute, Chinese Academy of Agricultural Sciences , Zhenjiang, Jiangsu Province , China
Bonetta Dario
Electronic publication date: 2021 Aug 6
Publication date: 2021
Volume: 9
Electronic Location ID: e11964
Received 2020 Dec 28; Accepted 2021 Jul 21
Copyright: © 2021 Chao et al.
Copyright year: 2021
Copyright holder: Chao et al.
License: This is an open access article distributed under the terms of the Creative Commons Attribution License, which permits unrestricted use, distribution, reproduction and adaptation in any medium and for any purpose provided that it is properly attributed. For attribution, the original author(s), title, publication source (PeerJ) and either DOI or URL of the article must be cited.
License URL: https://creativecommons.org/licenses/by/4.0/

Keywords: Gene family, Genome-wide, Lignin, Mulberry, Zinc stress

Funding: Crop Germplasm Resources Protection Project of the Ministry of Agriculture and Rural Affairs of the People’s Republic of China 19190172 National Infrastructure for Crop Germplasm Resources NICGR-2019-43 China Agriculture Research System CARS-18-ZJ0205 This work was jointly supported by the Crop Germplasm Resources Protection Project of the Ministry of Agriculture and Rural Affairs of the People’s Republic of China (19190172), the National Infrastructure for Crop Germplasm Resources (NICGR-2019-43), and the China Agriculture Research System (CARS-18-ZJ0205). The funders had no role in study design, data collection and analysis, decision to publish, or preparation of the manuscript.

==============================
Mulberry (Morus, Moraceae) is an important economic plant with nutritional, medicinal, and ecological values. Lignin in mulberry can affect the quality of forage and the saccharification efficiency of mulberry twigs. The availability of the Morus notabilis genome makes it possible to perform a systematic analysis of the genes encoding the 11 protein families specific to the lignin branch of the phenylpropanoid pathway, providing the core genes for the lignin toolbox in mulberry. We performed genome-wide screening, which was combined with de novo transcriptome data for Morus notabilis and Morus alba variety Fengchi, to identify putative members of the lignin gene families followed by phylogenetic and expression profile analyses. We focused on bona fide clade genes and their response to zinc stress were further distinguished based on expression profiles using RNA-seq and RT-qPCR. We finally identified 31 bona fide genes in Morus notabilis and 25 bona fide genes in Fengchi. The putative function of these bona fide genes was proposed, and a lignin toolbox that comprised 19 genes in mulberry was provided, which will be convenient for researchers to explore and modify the monolignol biosynthesis pathway in mulberry. We also observed changes in the expression of some of these lignin biosynthetic genes in response to stress caused by excess zinc in Fengchi and proposed that the enhanced lignin biosynthesis in lignified organs and inhibition of lignin biosynthesis in leaf is an important response to zinc stress in mulberry.

Introduction

Lignin is an important component of plant cell walls and has important functions in plant growth and stress resistance (Chun et al., 2019). In turn, owing to its recalcitrant nature and complexity, lignin limits the efficient conversion of lignocellulosic biomass to ethanol (Ragauskas et al., 2014; Zabed et al., 2016). The modification of trees with less lignin or with more-degradable lignin along with normal growth, which can improve the quality of forage and saccharification efficiency, has become a hot topic (Dixon, Reddy & Gallego-Giraldo, 2014; Umezawa, 2018).

The lignin biosynthesis pathway has been deciphered and revised since its discovery decades ago (Whetten & Sederoff, 1995). As of now, a total of 11 enzymes have been identified to play a role in monolignol biosynthesis (Zhao, 2016). The monolignol biosynthesis pathway generally refers to the branch of phenylpropanoid pathway starting with the deamination of phenylalanine and leading to the production of hydroxycinnamyl alcohols. The general phenylpropanoid pathway contains phenylalanine ammonia-lyase (PAL), cinnamate 4-hydroxylase (C4H) and 4-coumarate: CoA ligase (4CL) and provides hydroxycinnamoyl-CoA esters as precursors for a wide range of end products, including lignin, flavonoids, anthocyanins and condensed tannins. In the monolignol-specific biosynthesis pathway, hydroxycinnamoyl-CoA esters undergo successive hydroxylation and O-methylation of their aromatic rings, as well as redox reactions, to produce the monolignols (Zhao, 2016). Coumaroyl shikimate 3′-hydroxylase (C3′H) and ferulate 5-hydroxylase (F5H) are responsible for the hydroxylation process. Shikimate O-hydroxycinnamoyl -transferase (HCT), caffeoyl CoA 3-O-methyltransferase (CCoAOMT) and caffeate/5-hydroxyferulate O-methyltransferase (COMT) are involved in the O-methylation process. Caffeoyl shikimate esterase (CSE) was recently discovered to convert caffeoyl shikimate into caffeate and consists of a bypass with 4CL (Saleme et al., 2017; Vanholme et al., 2013). Redox reactions are catalyzed successively by cinnamoyl CoA reductase (CCR) and cinnamyl alcohol dehydrogenase (CAD) to achieve the conversion of the side-chain carboxyl to an alcohol group. CCR and CAD constitute the primary pathway for monolignol biosynthesis (Zhao, 2016).

Mulberry (Morus, Moraceae) is an important economic plant in Asia with considerable nutritional and medicinal values (Yuan & Zhao, 2017). Moraceae is one of the closest relatives of Rosaceae and mulberry diverged from Cannabis sativa (Cannabaceae) 63.5 Mya, from apple/strawberry (Rosaceae) 88.2 Mya and from Medicago truncatula (Fabales) 101.6 Mya (He et al., 2013; Jiao et al., 2020). Many studies have shown the great potential of this plant in the energy, food and pharmaceutical industries. Mulberry has long been cultivated for sericulture, which shaped the world’s history through the Silk-Road. Furthermore, a large number of by-products of branches twigs have been produced from the large-scale cultivation of mulberry trees in traditional sericulture, and mulberry has been gradually considered a potentially new energy plant providing biomass for the production of biofuels (Łochyńska, 2015; Tang, Liu & Chen, 2012). Studies of lignin biosynthesis have been widely reported for energy plants and forage plants, such as poplar, Medicago sativa L. and Eucalyptus grandis (Carocha et al., 2015; Hamberger et al., 2007; Lee et al., 2011; Shi et al., 2010). Recently, Wang et al. characterized four Ma4CL genes from M. atropurpurea cv. Jialing No. 40. and revealed the functional divergence of Ma4CL (Wang et al., 2016).

The availability of the Morus notabilis genome and an increasing number of transcriptomic data for mulberry allows comprehensive genome-wide analyses of lignin biosynthesis genes in this species (Li et al., 2014). In addition, a recent study has been reported to reveal the chromosome-level genome of Morus alba (Jiao et al., 2020). Genome-wide screening, combined with de novo transcriptome data, was performed in this study on Morus notabilis and Morus alba variety Fengchi to obtain the genes putatively included in the 11 monolignol gene families. M. alba is one of the most widely cultivated mulberry in China. M. alba variety Fengchi is a new variety created by Sericultural Research Institute, Chinese Academy of Agricultural Sciences, expected to spread and grow in extreme environment conditions and used as heavy metal hyperaccumulators and forage. A phylogenetic tree and expression profile were used to further identify the bona fide genes involved in lignin biosynthesis, and finally, we provided a lignin toolbox consisting of 19 genes in Morus notabilis and 17 genes in Morus alba variety Fengchi, which will be convenient for researchers to explore and modify the monolignol biosynthesis pathway in this genus. We also assessed the potential roles of lignin biosynthetic genes in response to stress caused by the excess of zinc in Fengchi and proposed that the promotion of lignification in lignifying organs, associated with the inhibition of lignin deposition in leaves, is an important response to zinc stress in mulberry.

Materials and Methods

Plant materials

The materials used in this study were obtained from the National Germplasm Resource Nursery of the Institute of Sericulture, Chinese Academy of Agricultural Sciences. Annual seedlings, Morus alba L. variety Fengchi, were transplanted into plastic flowerpots, and the potted plants were irrigated with 400 ml/kg of Murashige and Skoog (MS) medium to provide nutrients (Susheelamma et al., 1996). Zinc sulfate powder was applied near the roots of the mulberry trees as excess zinc stress treatment (450 mg/kg). Changes of proline and superoxide dismutase (SOD) concentration were determined on the 15th day (Fig. S1). The root, stem and leaf tissues were quickly frozen in liquid nitrogen and stored at −80 °C. This experiment was performed using three biological replicates. These collected samples were used for both RNA-seq and RT-qPCR (quantitative real-time PCR) analysis.

Genome-wide screening of candidate genes for the lignin toolbox in mulberry

Bona fide genes involved in lignin biosynthesis with functional characterization from different plants were collected as a query sequence for an HMMer search using MorusDB online (https://morus.swu.edu.cn/morusdb) (Li et al., 2014). The sequence identity (>45%), e-value (<e-100) and full score (>400) were used to screen for candidate gene family members. For some gene family members such as CSE and CCoAOMT, the thresholds were flexible to obtain as many as possible candidate genes. A local blastp search was also performed to identify the Selaginella moellendorffii SmF5H and Fengchi homologs (Camacho et al., 2009). All of the query sequences used and candidate genes obtained are available in Table S1.

D. novo transcriptome assembly of Morus alba variety Fengchi

Transcriptome de novo assembly was carried out with the short reads assembling program Trinity (Grabherr et al., 2011). Unigenes were aligned by BLASTx (e < 0.00001) to protein databases in nr, Swiss-Prot, KEGG and COG/KOG. The best alignment results were chosen to determine the sequence direction of unigenes. When a unigene could not be aligned to any of these protein databases, the protein coding sequence and sequence direction were confirmed using ESTscan (Iseli, Jongeneel & Bucher, 1999). The data set is available with accession number PRJNA660559 in the National Center for Biotechnology Information (NCBI).

Sequence alignment and phylogenetic analysis

Putative protein sequences of different plant species were used for alignment and phylogenetic analysis. Sequences used for phylogenetic analysis were screened from various sources based on the platform PLAZA 3.0 (http://bioinformatics.psb.ugent.be/plaza/). Sequences from the gymnosperm Picea sitchensis and the fern Selaginella moellendorffii were obtained using BlastP in the NCBI database. Bona fide lignin related genes in different plants were obtained based on published studies (Carocha et al., 2015; Raes et al., 2003). Alignment was performed using DNAman 8.0 (Lynnon BioSoft, San Ramon, CA, USA) with default parameters. Phylogenetic trees were constructed using Mega 7.0 with the maximum-likelihood method (Kumar, Stecher & Tamura, 2016). The phylogenetic tree was assessed by bootstrapping using 1000 bootstrap replicates and marked above nodes only if greater than 50. The JTT substitution model and G+I rates among sites model were selected as parameters for building the tree. The putative protein sequences used are listed in Table S2.

Expression profile analysis

The gene expression based on the large-scale transcriptome data was calculated and normalized to RPKM (reads per kb per million reads). Transcriptome data of different tissues and organs (root, bark, leaves, winter bud, male-flower) in Morus notabilis was obtained from Mrousdb (https://morus.swu.edu.cn/morusdb) (Li et al., 2014). RNA-seq data for Fengchi different organs (root, stem and leaf) was aligned to de-novo transcriptome assembly of Morus alba variety Fengchi using bowtie2 and RPKM values for unigenes were calculated using deptools v2.0 based on the bam files (Langdon, 2015; Ramirez et al., 2014). RT-qPCR (quantitative real-time PCR) was also performed to validate gene expressions of 23 bona fide clade genes in different organs and the change of their expression levels after zinc treatment using ABI StepOnePlus™ Real-Time PCR System (USA). Genes that showed preferential expression in lignifying tissue or organs (bark, root and stem) were considered as candidate lignin-related genes. Primers based on the coding sequences of these genes were designed using Primer-Blast. The primers are available in Table S3 and the melt curve of each gene is provided in Fig. S2. Actin was used as reference gene according to previous studies (Shukla et al., 2019). Tbtools was used to visualize the expression profile (Chen et al., 2020), and Graphpad Prism8.0 was used to visualize the RT-qPCR results. SPSS19.0 was used to perform T-test and ANOVA, p < 0.05 was marked as significant. Three biological replicates were considered for transcriptome data and two biological replicates with three technical replicates respectively were performed for RT-qPCR.

Results

Genome-wide screening of monolignol biosynthesis pathway-related genes

Morusdb (https://morus.swu.edu.cn/morusdb), which provides the genome and transcriptome information for Morus notabilis, was used to perform genome-wide screening of candidate genes involved in monolignol biosynthesis. Finally, we obtained 56 candidate genes based on the HMMer search and blastp results (Table S1). In addition, we identified their homologs in Fengchi, a Morus alba variety bred by our institute, based on our de novo transcriptome data. Most (49/56) of the corresponding homologs were identified in Fengchi using candidate genes in Morus notabilis as a reference sequence.

Phenylalanine ammonia-lyase (PAL)

PAL (EC: 4.3.1.5) catalyzes the deamination of phenylalanine to produce cinnamic acid and is the initial step in the general phenylpropanoid pathway. We constructed a phylogenetic tree (Fig. 1A) using both (Morus notabilis Mn) MnPAL and (Fengchi Fc) FcPAL and bona fide PAL data reported in other species. Seven MnPALs were identified and clustered as bona fide PALs. However, only three homologs, FcPAL3, FcPAL6 and FcPAL7, were found in Fengchi based on de novo transcriptome data. MnPAL7 and FcPAL7 were quite divergent compared with other PALs in angiosperms and are closer to PALs from gymnosperms, which is similar to EgrPAL2 in Eucalyptus grandis (Carocha et al., 2015). Both MnPAL7 and FcPAL7 showed a low expression level in various tissues and organs and exhibited no obvious preference in the lignified tissues and organs (Fig. 1B). MnPAL1, 2, 4, and 5 are phylogenetically close to AtPAL1 and AtPAL2, which have been reported to be mainly involved in anthocyanin production (Cochrane, Davin & Lewis, 2004; Huang et al., 2010). MnPAL1 and 5 were preferentially expressed in lignifying organs and tissues (root and bark), which differed from the expression patterns of MnPAL2 and 4 (Fig. 1B). MnPAL1, 2, 4, and 5 (L484_024371, L484_024373, L484_024372, L484_024369) have high sequence identity (Aligned protein sequence identity >99%) and are located close to each other in the genome, forming a gene cluster. Although MnPAL1, 2, 4 and 5 showed high expression levels in the studied organs and tissues in Morus notabilis, we could not find or distinguish homologs of MnPAL1, 2, 4, and 5 in Fengchi. Mn PAL3, 6 and FcPAL3, 6 are phylogenetically close to AtPAL4, 5 and PtrPAL4, 5, which are reported to express more specifically in xylem tissues (Raes et al., 2003). MnPAL3 and FcPAL3 also showed an expression preference in the root, stem or bark, with a high overall expression level, while MnPAL6 and FcPAL6 showed low overall expression levels in all of the examined organs and tissues (Fig. 1B). RT-qPCR results also validated the expression preference of FcPAL3 in stems (Fig. S3). Based on the above facts, MnPAL1, 3,5 and FcPAL3 are the PAL genes most likely to be involved in lignification.

Figure 1 Phylogenetic analysis and expression profile of PAL and 4CL gene family in mulberry.

(A) Phylogenetic analysis of PALs; (B) Expression profiles of PAL gene family in different tissues or organs in Morus notabilis and Fengchi; (C) Phylogenetic analysis of 4CLs; (D) Expression profiles of 4CL gene family in different tissues or organs in Morus notabilis and Fengchi. Red full circles indicating PALs or 4CLs from dicots, blue full circles indicating PALs or 4CLs from monocots, green full circles indicating PALs or 4CLs from gymnosperms and yellow full circles indicating PALs from ferns or moss. Putative protein sequences were used for phylogenetic analysis and the sequences information is available in Table S2. Mn indicating Morus notabilis and Fc indicating Fengchi. L0, leaf without Zinc treatment; S0, stem without zinc treatment; R0, root without zinc treatment. Bona fide clades were marked using different color shading.

4-Coumaric acid coenzyme A ligase (4CL)

4CL (EC: 6.2.1.12) belongs to the ANL (AMP-producing adenylating superfamily of enzymes) superfamily and catalyzes the formation of CoA thiol esters of 4-coumarate and other 4-hydroxycinnamates, which are important input metabolites, especially for lignin biosynthesis and flavonoid biosynthesis (Ehlting et al., 1999). The bona fide 4CL clade in angiosperm comprises three classes. Clade I contains 4CLs, which are mainly involved in lignin biosynthesis, including At4CL1, 3, 4, Pto4CL1, 3, 4, 5, Mn4CL1, 2, 4 and Fc4CL1, 2, 4 (Fig. 1C). The expression profiles of Mn4CL1, 2, 4 and Fc4CL1, 2, 4 also indicated preferential expression in the root, stem or bark (Fig. 1D and Fig. S3). Mn4CL3 and Fc4CL3 were clustered in Clade II together with At4CL3, Pto4CL2 and Os4CL2, which have been reported to be associated with flavonoid and soluble phenolic biosynthesis (Gui, Shen & Li, 2011; Li et al., 2015; Rao et al., 2015). Mn4CL3 showed a high expression level in male flowers but a low expression level in winter buds, consistent with its possible function in flavonoid biosynthesis. The third clade only contained Os4CLs (Os4CL1/3/4/5), which are thought to be distinct from the lignin-associated clade I 4CLs found in dicots. We also found that Mn4CL5, 6 and Fc4CL5, 6 were in a separate cluster and were phylogenetically close to (Plagiochasma appendiculatum) Pa4CL, the liverwort Plagiochasma appendiculatum. Mn4CL5 showed a similar expression pattern to Mn4CL3 and a high expression in male flowers and bark. Mn4CL6 had an expression specific to male flowers. These facts indicate that Mn4CL5, 6 may also be involved in flavonoid and soluble phenolic biosynthesis, given the high flavonoid content in mulberry. Mn4CL5, 6 and Fc4CL5, 6 are divergent from 4CLs in angiosperms and still need to be further studied to identify their roles in mulberry. Therefore, Mn4CL1, 2, 3, 4 and Fc4CL1, 2, 3, 4 are the 4CL genes most likely to be involved in lignification.

Hydroxylation steps in the general phenylpropanoid pathway

C4H (EC: 1.14.13.11) and C3′H (EC: 1.14.14.1) catalyze the hydroxylation steps. C4H and C3′H belong to the CYP73 and CYP98 families, respectively which are members of the cytochrome P450 monooxygenase superfamily. C4H is generally encoded by small gene family, except in Arabidopsis, which has only one AtC4H. Studies in Populus have shown a C4H–C3′H complex that more efficiently catalyzes hydroxylation steps (Chen et al., 2011). Here, we identified three candidate C4Hs in mulberry. MnC4H1, 2 and FcC4H1, 2 clustered with AtC4H and PoptrC4H1, 2 as Clade I, which is responsible for lignin biosynthesis (Fig. 2A). MnC4H1, 2 showed a high expression in all organs and tissues. FcC4H1, 2 was preferentially expressed in lignified organs (Fig. 2C, Fig. S3). MnC4H3 and FcC4H3 were grouped with PoptrC4H3 and are distinct from MnC4H1, 2 and FcC4H1, 2. Similar to PoptrC4H3, MnC4H3 and FcC4H3 had a very low expression level in all of the studied organs. Therefore, MnC4H1, 2 and FcC4H1, 2 are the C4H genes most likely to be involved in lignification.

Figure 2 Phylogenetic analysis and expression profile of C3′H and C4H gene families in mulberry.

(A) Phylogenetic analysis of C4Hs; (B) Phylogenetic analysis of C3′Hs. (C) Expression profiles of C3′H and C4H gene family in different tissues or organs in Morus notabilis and Fengchi. Red full circles indicating proteins from dicots, blue full circles indicating proteins from monocots, green full circles indicating proteins from gymnosperms and yellow full circles indicating proteins from ferns or moss. Bona fide clades were marked using different color shadings. Putative protein sequences were used for phylogenetic analysis and the sequences information is available in Table S2. Mn indicating Morus notabilis and Fc indicating Fengchi. L0, leaf without zinc treatment; S0, stem without zinc treatment; R0, root without zinc treatment.

Although C3′H was shown to catalyze the conversion of p-coumaric acid into caffeic acid in vitro, further studies demonstrated that its activity in vitro is the conversion of p-coumaroyl shikimate to caffeoyl shikimate (Abdulrazzak et al., 2006; Franke et al., 2002, Schoch et al., 2001). Based on our phylogenetic analysis, only MnC3′H1 and FcC3′H1 clustered with StC3′Hs as bona fide clade II (Fig. 2B). It is interesting to note that MnC3′H1 and FcC3H1 are phylogenetically closer to C3′Hs in monocots, other than C3′Hs such as AtC3′H and PoptrC3′H in dicots (Fig. 2B). Other candidates, including MnC3′H2-5 or FcC3′H2-5, is in a separate cluster without any C3′H orthologs in other plants. It seems that C3′H in mulberry is similar to that in A. thaliana, which also has only one C3′H (Raes et al., 2003). MnC3′H1showed a high expression in all of the studied organs, and FcC3′H1 was preferentially expressed in the stem, according to both transcriptome data and RT-qPCR, which likely involves lignification (Fig. 2C and Fig. S3). Therefore, MnC3′H1 and FcC’3H1 are the C3H genes most likely to be involved in lignification.

Hydroxycinnamoyl CoA: shikimate hydroxycinnamoyl transferase (HCT) and caffeoyl shikimate esterase (CSE)

HCT (EC: 2.3.1.133) combined with C3′H (p-coumarate 3-hydroxylase) catalyzes two steps to change the carbon flux from H to G and S lignin units. HCT belongs to the BAHD acyltransferase family and is able to utilize a variety of non-native substrates (Chiang et al., 2018; D’Auria, 2006). P-coumaroyl-CoA and caffeoyl-CoA are preferential substrates for HCTs and HCTs catalyze the acylation of CoA esters with shikimate, producing shikimate esters containing coumaric acid or caffeic acid. The reverse reaction for the formation of caffeoyl-CoA from caffeoyl shikimate is also catalyzed by HCT. Hydroxy-cinnamoyl CoA: quinate hydroxycinnamoyl transferase (HQT) is another acyl transferase that uses quinic acid instead of shikimic acid as the acceptor compound and is involved in chlorogenic acid biosynthesis, not lignin biosynthesis (Niggeweg, Michael & Martin, 2004). We constructed a phylogenetic tree using both HCTs and HQTs to distinguish bona fide HCT clades (Fig. 3A). Six candidate MnHCTs were grouped as bona fide HCTs with HCTs in angiosperms. MnHCT2, 3 (L484_000457, L484_018078) and MnHCT5, 6 (L484_017530, L484_017529) had high sequence similarity (aligned protein sequence identity >95%). We could not distinguish FcHCT2, 3 and FcHCT5, 6 based only on transcripts; therefore, we named FcHCT2 and 5 based on their similar expression pattern to MnHCT2 and MnHCT5. Among all MnHCTs, MnHCT1, 2 and FcHCT1, 2 were preferentially expressed in lignified organs and tissues (stems, roots and bark) and are likely involved in monolignol biosynthesis (Fig. 3C, Fig. S3). Other MnHCTs and FcHCTs showed relatively low expression in all organs and tissues. MnHCT5, 6 and FcHCT5 are phylogenetically divergent from other MnHCTs and MnHCTs and showed preferential expression in the leaf (FcHCT5) and in the winter bud and leaf (MnHCT5), which indicates their possible different roles as opposed to lignification in mulberry. Therefore, MnHCT1, 2 and FcHCT1, 2 are the HCT genes most likely to be involved in lignification.

Figure 3 Phylogenetic analysis and expression profile of HCT and CSE gene families in mulberry.

(A) Phylogenetic analysis of HCTs; (B) Phylogenetic analysis of CSEs; (C) Expression profiles of HCT and CSE gene family in different tissues or organs in Morus notabilis and Fengchi. Red full circles indicating proteins from dicots, red empty circles indicating HQTs, blue full circles indicating proteins from monocots, green full circles indicating proteins from gymnosperms and yellow full circles indicating proteins from ferns or moss. Bona fide clades were marked using different color shadings. Putative protein sequences were used for phylogenetic analysis and the sequences information is available in Table S2. Mn indicating Morus notabilis and Fc indicating Fengchi. L0, leaf without zinc treatment; S0, stem without zinc treatment; R0, root without zinc treatment.

AtCSE was first characterized as lysoPL2, a member of the monoacylglycerol lipase (MAGL) gene family in Arabidopsis (Kim et al., 2016; Gao et al., 2010). AtCSE was first reported as caffeoyl shikimate esterase by Vanholme et al. (2013) in Arabidopsis because of its ability to convert caffeoyl shikimate into caffeate. Further analysis of an A. thaliana cse-2 (caffeoyl shikimate esterase 2) knockout mutant that presented a reduced lignin content enriched in H units and depleted in S units indicated the involvement of CSE (EC: 3.1.1.) in lignin biosynthesis (Vanholme et al., 2013). CSE competes with HCT for the substrate caffeoyl shikimate. MnCSE1 and FcCSE1 are phylogenetically close to AtCSE, PoptrCSE1,2 and MtCSE, which have been reported to be involved in lignin biosynthesis (Fig. 3B) (Ha et al., 2016; Saleme et al., 2017). In addition, FcCSE1 showed preferential expression in lignified organs and tissues (Fig. 3C, Fig. S3). MnCSE2 and FcCSE2 showed close relationship with AtGAML1 which was reported to harbor MAG lipase activities and lysophosphatidylcholine (LPC) and/or lysophosphatidy -lethanolamine (LPE) hydrolase activities. MnCSE3, without a homolog in Fengchi, was far from the bona fide CSEs and cluster with AtMAGL9 and 12. MnCSE3 had an expression preference in winter buds and male flowers. In general, MnCSE1 and FcCSE1 are lignin-related CSEs, but MnCSE2 and FcCSE2 are monoacylglycerol lipase.

The methylation steps

COMT (EC: 2.1.1.68) and CCoAOMT (EC: 2.1.1.104) are both involved in the methylation steps of the monolignol pathway (Zhong et al., 1998). CCoAOMT catalyzes the methylation of caffeoyl CoA to produce feruloyl CoA and is reported to be responsible for G and S-type lignin. Only one CCoAOMT in mulberry, MnCCoAOMT1 or FcCCoAOMT1, clusters in the bona fide clade with AtCCoAOMT and PtoCCoAOMT1 and 2 (Fig. 4A). Both MnCCoAOMT1 and FcCCoAOMT1 showed a high expression level in the lignified organs (Fig. 4C, Fig. S3). FcCCoAOMT1 had the highest expression level in the stems, about 50-fold higher than that in the leaves (Fig. 4C). MnCCoAOMT1 had high expression in the root, bark and male flowers, with the highest expression in the root (two-fold higher than the expression in the bark or male flower, five-fold higher than the expression in the leaf). Two other candidates, MnCCoAOMT2, 3 and FcCCoAOMT2, 3 belong to the CCoAOMT-like clade and had a low expression level in all organs and tissues. Therefore, MnCCoAOMT1 and FcCCoAOMT1 are the CCoAOMT1 genes most likely to be involved in lignification.

Figure 4 Phylogenetic analysis and expression profile of CCoAOMT and COMT gene families in mulberry.

(A) Phylogenetic analysis of CCoAOMTs; (B) Phylogenetic analysis of COMTs; (C) Expression profiles of CCoAOMT and COMT gene family in different tissues or organs in Morus notabilis and Fengchi. Red full circles indicating proteins from dicots, blue full circles indicating proteins from monocots, green full circles indicating proteins from gymnosperms and yellow full circles indicating proteins from ferns or moss. Bona fide clades were marked using different color shadings. Putative protein sequences were used for phylogenetic analysis and the sequences information is available in Table S2. Mn indicating Morus notabilis and Fc indicating Fengchi. L0, leaf without zinc treatment; S0, stem without zinc treatment; R0, root without zinc treatment.

In angiosperms, COMT (EC: 2.1.1.68) is involved in the synthesis S precursors and is now considered to be primarily involved in the synthesis of S units through the preferential methylation of 5-hydroxyconiferyl aldehyde into sinapaldehyde based on functional analysis in several species (Davin et al., 2008) In mulberry, only one COMT, MnCOMT4 or FcCOMT4 was identified as a bona fide COMT together with AtCOMT and PoptrCOMT1,2, based on our phylogenetic analysis (Fig. 4B). MnCOMT4 and FcCOMT4 showed obvious expression preference in the lignified organs and tissues (Fig. 4C, Fig. S3) and should be responsible for lignin biosynthesis in mulberry. Other candidate MnCOMTs and FcCOMTs were in a separate cluster and phylogenetically far from the bona fide clade. MnCOMT2, 5 showed a relatively high expression in male flowers compared with that in the leaf, winter bud and bark. MnCOMT1 and 6 showed a very low expression in all of the detected organs, and the RPKM of MnCOMT6 based on the published transcriptome data is not available. FcCOMT1, 2, 3, 5, and 6 had similar expression patterns, with overall low expression in all organs and a relatively high expression in roots compared with the stem and leaf. These COMT-like genes need more evidence and functional analysis for elucidating their roles in mulberry. Therefore, MnCOMT4 and FcCOMT4 are the COMT genes most likely to be involved in lignification.

Hydroxylation step specific for S lignin production

F5H (EC:1.14.13) belongs to the CYP84 family and is similar to C3H and C4H as a member of the cytochrome P450 monooxygenases. F5H (EC: 1.14.13), also called CAld5H because of its substrate preference for coniferaldehyde/coniferyl alcohol (Humphreys, Hemm & Chapple, 1999), catalyzes the hydroxylation step specific for the production of sinapyl alcohol and, ultimately, S lignin. The discovery of SmF5H in the lycophyte Selaginella moellendorffii revealed a novel P450 (CYP788A1) (Weng et al., 2008). SmF5H shares only 37% amino acid sequence identity with its angiosperm counterparts and can also use p-coumaraldehyde and p-coumaryl alcohol as substrates to efficiently produce caffeoyl aldehyde and caffeoyl alcohol. Therefore, SmF5Hs can divert G-substituted intermediates toward S lignin synthesis through related but distinct pathways compared with angiosperms (Weng & Chapple, 2010). In addition to the genome-wide screening using F5Hs from angiosperms, we carried out blastp using SmF5H as a query to find more F5H-like sequences MnF5H1, 2 and FcF5H1, 2 were identified as candidate F5H based on a Hmmer search using F5Hs from angiosperms. MnF5H1 and FcF5H1 clustered with AtF5H and PoptrF5H1, 2 and belong to the bona fide clade in angiosperms (Fig. 5A). MnF5H1 and FcF5H1 showed obvious expression preference in lignified organs and tissues and are likely to be involved in lignin biosynthesis (Fig. 5B, Fig. S3). In contrast, MnF5H2 and FcF5H2 were far from the bona fide clade and had very low expressions in all organs and tissues. Other candidate F5Hs named MnF5H3(Sm), MnF5H4(Sm) or FcF5H3(Sm), FcF5H4(Sm) were identified, sharing about 45% protein sequence identity with SmF5H. MnF5H3(Sm), MnF5H4(Sm) and FcF5H3(Sm), FcF5H4(Sm) showed relatively high expression in the root, and MnF5H3(Sm) also had a high overall expression in the bark, male flowers and leaf. These SmF5H-like proteins in mulberry may be involved in the response to zinc stress since FcF5H3(Sm) and FcF5H4(Sm) both showed a decreased expression in the leaf after zinc treatment (Fig. S4). Therefore, MnF5H1 and FcF5H1 are the F5H genes most likely to be involved in lignification.

Figure 5 Phylogenetic analysis and expression profile of F5H gene family in mulberry.

(A) Phylogenetic analysis of F5Hs; (B) Expression profiles of F5H gene family in different tissues or organs in Morus notabilis and Fengchi. Red full circles indicating F5Hs from dicots, blue full circles indicating F5Hs from monocots, green full circles indicating F5Hs from gymnosperms and yellow full circles indicating F5Hs from ferns or moss. Bona fide clades were marked using different color shadings. Putative protein sequences were used for phylogenetic analysis and the sequences information is available in Table S2. Mn indicating Morus notabilis and Fc indicating Fengchi. L0, leaf without zinc treatment; S0, stem without zinc treatment; R0, root without zinc treatment.

The last two reductive steps

CCR (EC: 1.2.1.44) is the first committed enzyme for a specific branch of monolignol biosynthesis and converts various cinnamoyl-CoA esters (p-coumaroyl-CoA, caffeoyl-CoA, feruloyl-CoA and sinapoyl-CoA) to produce their corresponding hydroxycinnamaldehydes, which are further reduced into different monolignols by another reductase called cinnamyl-alcohol dehydrogenase (CAD EC: 1.1.1.195). CCR and CAD are involved in the primary pathway of monolignol biosynthesis. A recent study showed that PoptrCAD1 and PoptrCCR2 can form a complex to regulate monolignol biosynthesis in Populus (Yan et al., 2019).

Six candidate CCRs from the mulberry genome were screened; phylogenetic analysis showed that MnCCR1, 2 and FcCCR1, 2 belonged to the bona fide clade with AtCCR1,2, MtCCR1,2 and PtoCCR1,7 (Fig. 6A). Further motif-aware analysis based on our previously reported workflow further validated that MnCCR1,2 and FcCCR1,2 belonged to bona fide CCRs and other candidate CCRs should be CCR-like (Fig. S5) (Chao et al., 2019). Similar to AtCCR1, 2 and PtoCCR1, 7, different bona fide CCRs in mulberry also had different expression patterns. MnCCR1 and FcCCR1 had a high overall expression, with the highest expression level in the bark or stem, while MnCCR2 and FcCCR2 had quite low expression in all organs and tissues (Fig. 6C, Fig. S3). Therefore, MnCCR1 and FcCCR1 are likely to play a predominant role in monolignol biosynthesis. As both phylogenetic analysis and motif-aware analysis showed, other MnCCRs and FcCCRs belong to the CCR-like cluster with unknown functions.

Figure 6 Phylogenetic analysis and expression profile of CCR and CAD gene families in mulberry.

(A) Phylogenetic analysis of CCRs; (B) Phylogenetic analysis of CADs; (C) Expression profiles of CCR and CAD gene family in different tissues or organs in Morus notabilis and Fengchi. Red full circles indicating proteins from dicots, blue full circles indicating proteins from monocots, green full circles indicating proteins from gymnosperms and yellow full circles indicating proteins from ferns or moss. Bona fide clades were marked using different color shadings. Putative protein sequences were used for phylogenetic analysis and the sequences information is available in Table S2. Mn indicating Morus notabilis and Fc indicating Fengchi. L0, leaf without zinc treatment; S0, stem without zinc treatment; R0, root without zinc treatment.

CAD is the last enzyme in monolignol biosynthesis and uses various phenylpropenyl aldehyde derivatives as substrates to ensure the diversity of lignin. We obtained six candidate MnCADs, and the corresponding homologs in Fengchi were found except MnCAD4. FcCAD4 was different from all six candidate MnCADs, with high (73.88%) protein sequence identity to AtCAD1 which was reported to have very low catalytic activity in vitro and play roles in lignification of elongating stems (Eudes et al., 2006; Kim et al., 2004). MnCAD1, 2, 3, 4, 5 and FcCAD1, 2, 3, 5 belong to bona fide clades (Fig. 5B). MnCAD3, 4 and FcCAD3 clustered with AtCAD4, 5, PtoCAD1, and BdCAD5, which have been reported to be involved in lignin biosynthesis. Although the expression data for MnCAD3, 4 were not available in Morusdb, FcCAD3 showed expression specific to lignified organs based on our transcriptome data and RT-qPCR (Fig. 5C, Fig. S3). In addition to the above bona fide CADs, there is another kind of bona fide CAD with the present PtrSAD in Populus (Li et al., 2001). This PtrSAD has been reported to prefer sinapaldehyde as a substrate,however, our previous study on PtoCAD2 showed no obvious substrate preference (Chao et al., 2014). MnCAD1,2 and FcCAD1, 2 are phylogenetically close to the so-called PtrSAD and PtoCAD2 and cluster as another bona fide clade. MnCAD1 showed no obvious expression preference while FcCAD1 exhibit a preference for lignified organs based on RT-qPCR results (Fig. 5C, Fig. S3). MnCAD2 and FcCAD2 showed an expression preference for winter-bud or leaf (Fig. 5C, Fig. S3).

Summary of the lignin toolbox for mulberry and the response to zinc ion stress

Genes considered the lignin toolbox in mulberry were annotated in Table S4. Hierarchical clustering depicted a similar expression pattern for bona fide lignin biosynthetic genes we described above, which differed from that of genes identified as ‘like’ genes (candidate genes excluded from bona fide clade) (Fig. S6A). It was obvious that the bona fide genes had a higher overall expression in the studied organs than the ‘like’ genes. 21 of total 31 bona fide genes in mulberry can be classified as two main clusters based on their expression patterns. Cluster I (indicated as a blue star r) includes genes with obvious expression preferences in lignified organs such as stem and bark, and cluster II (indicated as a red star) includes genes with a high expression but no obvious preference in lignified organs and tissues. RT-qPCR also validated the expression preference in lignified organs for the lignin-related genes comprise the lignin toolbox for mulberry (Fig. S6B, Fig. S3 and Table S4).

Mulberry has been reported as heavy metal hyperaccumulators. Our results showed that lignin-related genes play important roles in responding to zinc stress. All we detected bona fide clade genes (22/23) including 16 core genes in lignin toolbox in Fengchi show significant expression change in at least one organ (Fig. 7A). Only FcCCoAOMT1showed no significant change in any detected organs after zinc excess treatment (Fig. 7A, Fig. S3, Table S4). Monolignol biosynthesis pathway in Fengchi showed overall up-regulation in root and stem but down-regulation in leaf (Fig. 7B). Most of these bona fide clade genes (13/22) were down-regulated in leaves. Five genes showed up-regulation in both leaf and lignified organs (Fig. 7C). It is likely that the promotion of lignin biosynthesis in lignified organs in mulberry is an important way to respond to zinc stress.

Figure 7 Expression change of bona fide clade genes in response to excess zinc stress in mulberry.

(A) Fold change of expression levels of 23 bona fide genes in Fengchi after excess zinc treatment; (B) Overall change of monolignol pathway in different organs after excess zinc treatment in Fengchi; (C) Clustering of 23 bona fide clade genes expression pattern in response to Zinc stress. Two biological replicates with three technical replicates respectively were performed for qRT-PCR. P-value was calculated using SPSS 19.0. An asterisk (*) indicates 0.01 < p <0.05; two asterisks (**) indicates 0.001 < p < 0.01 and three asterisks (***) indicates p < 0.001.

Discussion

Lignification toolbox in mulberry

Genes involving in secondary metabolism in mulberry were reported to have faster evolutionary rate (Jiao et al., 2020). Lignin biosynthesis is important pathway in land plants. Genome-wide screening of candidate genes involving in monolignol biosynthesis was performed here in mulberry. In total, 31 bona fide clade genes were obtained in Morus notabilis based on phylogenetic analysis, and 25 bona fide homologs were found in Fengchi, similar to Populus (25) and E. grandis (38) (Table S4) (Carocha et al., 2015; Shi et al., 2010). The loss of MnHCT and MnPAL homologs in Fengchi resulted in the above-mentioned change in the total numbers. The bona fide genes described above had a similar expression pattern and a higher overall expression in the studied organs compared with genes identified as ‘like’ genes. Combined with the expression profile in different organs of Morus notabilis and Fengchi, a total of 19 genes were identified as bona fide lignification-related genes, which is similar to that in E. grandis (17). These 19 genes were preferentially expressed in the lignified organs and tissues and probably represent the core lignification toolbox in mulberry (Fig. S6B, Fig. S3, Table S4).

The lignin biosynthesis pathway plays an important role in the response to stress caused by excess zinc in mulberry

Zinc is a trace element that is necessary for a healthy immune system and is important for people to maintain their fitness level. Studies have shown that dietary zinc can act as sleep modulator and is necessary for brain development and function (Cherasse & Urade 2017; Hambidge, 2000). Mulberry is a woody plant with resistance to zinc ions, and both leaves and fruits of mulberry are known as sites rich in zinc (Jiang et al., 2017; Srivastava et al. 2006). Black mulberry (Morus nigra) juice has high amounts of zinc and iron, which could help to improve the micronutrient status of pregnant women and children (Khalid, Fawad & Ahmed, 2011) A deficiency or excess zinc leads to oxidative stress. Moreover, the Zn-deficiency leads to abnormal development of leaves in mulberry (Kumar Tewari, Kumar & Nand Sharma, 2008).

Mulberry is able to uptake the heavy metal and was reported to immigrate 254,532.8 mg Zn every square meter plough layer soil (Jiang et al., 2017). The contents of zinc in different mulberry organs (leaf, root, bark and stem) are greatly different (Jiang et al., 2017). After excess zinc treatment, 23 core genes involved in lignin biosynthesis except CCoAOMT1 showed obvious expression changes in different organs (Fig. 7). Monolignol biosynthesis pathway in Fengchi showed overall up-regulation in root and stem but down-regulation in leaf (Fig. 7B). Relatively high expression of lignin related genes (total 24 genes) was also reported in response to zinc exposure in roots of Thlaspi caerulescens, one of the natural zinc hyperaccumulator species (Van de Mortel et al., 2006). Lignin has been reported to act as a metal-absorbing matrix in response to metal stress (Bhardwaj et al., 2014). It is likely that the promotion of lignin biosynthesis in lignified organs association with inhibition of lignin biosynthesis in leaf in mulberry is an important response to zinc stress. A similar situation was reported for Lens Culinaris and Phaseolus Mungo subjected to lead stress (Haider & Azmat, 2012).

Supplemental Information

Supplemental Information 1 Pro concentration and SOD concentration in mulberry leaves, roots and stems at 15 days after treatment.

Click here for additional data file.

Supplemental Information 2 The melt curve of each gene for RT-qPCR experiments.

Click here for additional data file.

Supplemental Information 3 qRT-PCR results for 23 bona fide clade genes in Fengchi.

Click here for additional data file.

Supplemental Information 4 qRT-PCR for FcF5H3(Sm) and FcF5H4(Sm) in Fengchi.

Click here for additional data file.

Supplemental Information 5 Alignment and motif analysis of CCRs. The motifs were marked in red box.

The motifs were marked in red box.

Click here for additional data file.

Supplemental Information 6 Expression profile of all candidate genes in mulberry.

(A) Hierarchical clustering of expression profiles of 56 candidate genes based on transcriptome data in mulberry; Blue star indicated the cluster I and red star indicate the cluster II. Red full circles indicated the bona fide clade genes (B). Expression profiles of 23 bona fide clade genes in Fengchi.

Click here for additional data file.

Supplemental Information 7 Genome-wide screening of monolignon biosynthesis pathway-related genes.

Click here for additional data file.

Supplemental Information 8 Sequence used for lignin-related gene analysis in mulberry.

Click here for additional data file.

Supplemental Information 9 Primers used for qRT-PCR.

Click here for additional data file.

Supplemental Information 10 Putative functional classification of 56 candidate lignification-related genes.

Click here for additional data file.

Supplemental Information 11 Raw data for qRT-PCR.

Click here for additional data file.

Supplemental Information 12 Basic annotation of all unigenes.

Click here for additional data file.

Additional Information and Declarations

Competing Interests

Author Contributions

DNA Deposition

Data Availability

The authors declare that they have no competing interests.

Nan Chao conceived and designed the experiments, performed the experiments, analyzed the data, prepared figures and/or tables, authored or reviewed drafts of the paper, and approved the final draft.

Ting Yu performed the experiments, analyzed the data, prepared figures and/or tables, and approved the final draft.

Chong Hou analyzed the data, authored or reviewed drafts of the paper, and approved the final draft.

Li Liu conceived and designed the experiments, authored or reviewed drafts of the paper, and approved the final draft.

Lin Zhang performed the experiments, authored or reviewed drafts of the paper, and approved the final draft.

The following information was supplied regarding the deposition of DNA sequences:

The data are available at NCBI SRA: PRJNA660559.

The following information was supplied regarding data availability:

The data are available at NCBI SRA: PRJNA660559.

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
