# Peer review of "Genome-wide analysis of the lignin toolbox for morus and the roles of lignin related genes in response to zinc stress"

_PeerJ, doi:10.7717/peerj.11964_

## Round 0.1 · original submission · Major Revisions

Both reviewers have made excellent suggestions to improve the quality of the manuscript. They have also made comments regarding clarifications required for the interpretation of the findings. I agree with the second reviewer that some experimental details need to be ironed out before the paper can be published.

·

Basic reporting

Most of the text is well-written, with no grammar mistakes, but some sentences are poorly written. For such sentences, I made suggestions that might help the authors.

I think the authors should cite more studies reporting the determination of the lignin toolbox in other species. More references on some key aspects of Morus (such as zinc stress in such species) might also be useful.

The structure of the article is acceptable. However, I have the suggestion to organize the gene families according to their order in the biosynthetic pathway. Currently, it may confound the reader. The figures are of good quality, but sometimes there are too much information on the phylogenetic trees, so the reader cannot depict the genes/proteins in the terminals.

Because the authors consider both species as "morus", sometimes the article becomes confusing. For example, when determining the toolbox based on the hierarchical clustering, I don't think considering all genes as from the same species is correct and this leads to wrong conclusions.

Experimental design

The research is original, as this is the first attempt to determine the lignin toolbox in mulberry.

Although the research question is well-defined, I think the authors should clarify their criteria to select their target genes. This is very confusing throughout the manuscript. The first part of the results section should present all the criteria and what would be consider a "good candidate". The authors should also include a conclusive sentence at the end of each section (i.e. for each gene family).

One major concern regarding the experimental design: the RNAseq data exploited to survey the lignin biosynthetic genes in M. notabilis do not include xylem tissues (or, at least, stem tissue), which are considered a hotspot for lignin deposition in woody species such as those of the Morus genus. As the authors did not provide detailed information on the tissues samples used for RNAseq and RT-qPCR analyses, it is hard to correlate gene expression with lignin deposition, which makes their criteria weaker. In the case of M. notabilis, the authors should have include another step within their pipeline to identify the lignin toolbox: RT-qPCR analyses in tissues contrasting for lignin deposition. This becomes obvious when analyzing the CAD family, for which expression data are not avaliable for the two most important candidate genes.

Methods are not described with sufficient details. Please see my comments below.

Validity of the findings

The selection of candidate genes for the lignin toolbox in mulberry is not completely reliable. While pertinent for some gene families, for others the expression pattern of some selected genes is not consistent with lignification. More importantly, the search for SmF5H homologues in mulberry does not make sense, as this gene belong to a clade that especifically evolved in Selaginella. The lack of a tissue considered as a "reference tissue" for lignification in their dataset for M. notabilis (i.e. xylem tissues) is another drawback of their exeprimental design.

Another concern is that the authors aim to determine the "role of lignin-related genes in zinc stress". However, they only analyzed the expression of the so-called lignin toolbox, which was determined based on developmental lignification. It is known that plants may employ different paralogues for each gene family when comparing developmental versus stress lignification. Therefore, to really understand this response, the authors should have analyzed all 49 genes found for the species.

Additional comments

- Line 18: it is important to explain what “ramulus mori” is
- Line 24: the sentence “Genes in the bona fide clade were focused on” is confusing; did you mean that you have focused on bona fide genes?
- Line 28: the process of naming your target species is a bit confusing; in the abstract, the reader can find Mulberry, ramulus mori, Morus notabilis, Morus alba variety Fengchi and morus; this problem is particularly relevant when you claim that the lignin toolbox was determined as 19 genes in morus; does the expression “morus” refer to species in the Morus genus? Wouldn’t be better to use “mulberry” as the general name?
- Line 45-27: well, this is actually not true; the monolignol biosynthetic pathway refers to the branch of the phenylpropanoid pathway leading to the synthesis of hydroxycinnamyl alcohols
- Line 53: the abbreviation for p-coumarate 3´-hydroxylase is C3´H (this is important because the recently characterized ascorbate peroxidase – APX – showing the ability to 3´-hydroxylate p-coumarate to caffeic acid is named C3H -> without the apostrophe)
- Line 53-55: please be careful with chemical nomenclature: 5´-hydroxylase and the “O” should be in italic in the name of the transferases
- Line 60-61: I understand that CCR and CAD are monolignol-specific enzymes, but why is 4CL included in this sentence as the “primary pathway for monolignol biosynthesis”?
- Line 66: please, explain here what “ramulus mori” is
- Line 70: with this sentence, are the authors referring to other studies reporting “lignin toolbox” for other plant species? Because if yes, these references are not appropriate
- Line 78: please explain why did you include M. alba variety Fengchi in this study; why was it important?
- Line 83-86: it is also not clear why did the authors decide to study the stress caused by excess of zinc; is this particularly important for mulberry production? Why not any other abiotic stress?
- Line 92-96: about the zinc treatment, is there a reference for this specific experimental design? I have several questions: i) why 450 mg/kg? ii) samples were harvested only at T=0 and after 15 days of treatment? Iii) why was the zinc stress analyzed only in Fengchi? iv) how old exactly the seedlings were? One year old? Why to analyze this in young seedlings? v) when the authors mentioned “three biological replicates”, did they mean only three individual plants per treatment?
- Line 103: what would be “SmF5H”?
- Line 112: in which database your data was deposited?
- Line 121: I suggest “The gene expression based on the large-scale transcriptome data was calculated…”
- Line 122-123: by realizing that RNAseq data are available for M. notabilis, it is even more important to explain why Fengchi was included in this study
- Line 122-123: the authors should provide detailed information on the tissue samples used for RNAseq and RT-qPCR analyses; were these tissues from adult plants? how old were the plants? In the case of roots, were they mature and lignified?
- Line 136: “monolignin” is not a correct term; please use either monolignol biosynthesis pathway or lignin biosynthesis pathway
- Line 141: here we have the first relevant information on Fengchi; but not only this is not enough but also is misplaced in the manuscript
- Line 146: the deamination of Phe into cinnamic acid is the initial step of the phenylpropanoid pathway in every plant species, not only woody plants; the only remark on this topic is that grasses harbor PALs that are able to use tyrosine to produce p-coumaric acid, by passing the reactions catalyzing by PAL + C4H
- Line 147: the authors should be consistent throughout the manuscript: the heatmap indicating gene expression data on PAL genes is represented in a different way when compared to the other gene families (genes in rows, not in columns)
- Line 147: I’m not sure that it is a good idea to adopt the abbreviation for the Fengchi genes (FcXXX) as it is; I mean, this is a variety but not a species, so your method is not really following what has been done for all other species; this can cause confusion in the future
- Line 156: the sentence “MnPAL1 and 5 showed an expression preference in the lignification of organs and tissues” does not make sense; are the authors trying to claim that the expression of such genes is preferential in these tissues?
- Line 156: also, the authors should make it clear to the reader which is the expression pattern expected for a gene to be involved in lignification in their panel of tissues; in this regard, my criticism is that the authors didn’t include “xylem tissues” in their analyses, which would be the best target for identifying lignin-related genes
- Line 168-169: why MnPAL2 and MnPAL5 are not considered strong candidates as well?
- Line 175-178: there is wrong information in the text; At4CL3 is the divergent one (Clade II), not At4CL2
- Line 176-178: the expression pattern of MnPAL4 is not consistent with lignification; the authors shouldn’t pool the description of all genes together because this is misleading
- Line 190: I suggest the authors include a conclusive sentence at the end of each section reporting the selected candidate genes to be part of the lignin toolbox in mulberry (as they did for PAL)
- Line 191: I suggest the authors to organize the description of each gene family according to their sequence in the pathway, not according to the type of reaction they perform; otherwise, it can get a little bit messy (and confusing for readers not familiar with the lignin biosynthetic pathway)
- Line 191: I have several comments for the section on the “hydroxylation steps”; 1) C3´H is not part of the general phenylpropanoid pathway (which consists of only PAL, C4H and 4CL); 2) C4H and C3H are not responsible for the production of all hydroxycinnamates, but ONLY p-coumarate (C4H) and caffeic acid* (*if C3H = APX, not CYP98); 3) the authors seem to confound C3´H (CYP98A3) with the recently characterized C3H (which is an ascorbate peroxidase); C3´H (CYP98A3) is responsible for the 3´-hydroxylation of p-coumaroyl shikimate into caffeoyl shikimate; in poplar, the protein complex made of C4H and C3´H was reported to catalyze the hydroxylation of p-coumarate to caffeic acid (Chen et al, 2011; doi: 10.1073/pnas.1116416109), but this has not been reported for any other plant species; C3H/APX is a bifunctional ascorbate peroxidase that not only uses ascorbate as substrate but also has the ability to convert p-coumarate into caffeic acid (Barros et al., 2019; doi: 10.1038/s41467-019-10082-7)
- Line 207: from which species is StC3H? my suggestion would be to include always the same plant species, for all gene families, for which a good amount of lignin biosynthetic genes have been functionally characterized
- Line 217: another mistake: the shikimate shunt, stated by the activity of HCT, channel the carbon flow away from H and towards the biosynthesis of G and S
- Line 219: HCT catalyze the acylation of CoA esters with shikimate, producing shikimate esters; please correct
- Line 243: why so many CSEs in the corresponding phylogenetic tree? Again, I suggest the authors to use the same plant species for all gene families, for organization reasons; the exception should be in cases it is particularly important to include additional proteins (such as HCT versus HQT)
- Line 244: regarding the CSE phylogeneny, previous studies have demonstrated that what we call a bona fide CSE with a function in lignification might be the result of a neofunctionalization within the monoacylglycerol lipase family (Ha et al., 2016; Kim et al, 2016; doi: 10.1111/tpj.13146); all other CSE-like genes are most likely MAGs; please, consider this in your phylogenetic analysis and when suggesting a function for CSE-like genes in lignification
- Lines 253: CCoAOMT is important for the synthesis of both G and S units
- Line 255: in figure 4, several CCoAOMT names in Clade I are overlapping, so it’s impossible to know which protein is which
- Line 281: please correct this sentence (remove “5-hydroxylas”)
- Line 289-290: in my opinion, it does not make sense to search for putative homologues of SmF5H in Angiosperms and, more importantly, to consider this strategy for this paper; in the original paper reporting the discovery of this gene (Weng et al., 2008; doi: 10.1073/pnas.0801696105) the authors performed phylogenetic analyses and concluded that SmF5H “belongs to a unique clade of Selaginella P450s that is distinct from all of the known P450s, suggesting an independent origin of F5H in Selaginella.”; so, the more closely related genes to SmF5H might be merely members of the more closely related CYP98 and CYP75 families
- Line 307: again, this remark on 4CL should be removed
- Line 312-314: please explain why bona fide CCRs should be considered CCR-like (this statement is contradictory, as a CCR-like gene necessarily indicates that it is not a bona fide gene); I believe the sentence is grammatically wrong
- Line 319: what is exactly “motif-aware analysis” and where can I find its corresponding results?
- Line 321-322: why exactly CAD is responsible for “the diversity of monolignols”? This enzyme catalyzes the same reaction for all lignin subunits, can could it contribute to the chemical diversity of such compounds?
- Line 324-326: this sentence is confusing; which enzyme was reported to be a CAD-like without detectable CAD activity in vitro?
- Line 328-329: because no expression data for MnCAD3 and 4 are available in Morusdb and no further expression analysis was performed, there is no further support for their role in lignification apart from the phylogenetic analysis; that’s why the authors should have provided RT-qPCR data also for M. notabilis
- Line 336: I don’t think the expression data support the role of FcCAD1 in lignification
- Line 340-343: the problem here is that you didn’t indicate which genes are “=-like” genes for each family; for some families (such as CCR), this is mentioned in the text, but the name of the genes should include this classification, so the reader can follow the story; first, you should describe what were the criteria to classify a gene as bona fide and as “-like”, then indicate in the text and in the figures, which genes belong to each classification (by just using “-like” in the abbreviation of each gene)
- Line 343: what should the reader understand with “(27/31)”? does it refer to the number of bona fide genes in each of the two Morus species?
- Line 341-348: regarding fig. S4, I think it is a great idea to perform hierarchical clustering for all the genes to further evaluate whether the selected toolbox indeed show a similar expression pattern (and, thus, cluster together); however, I’m confused by the fact that the authors apparently considered the orthologous genes in the two different species of morus as the same gene in this analysis; moreover, then they claim that RT-qPCR analysis “validated the expression preference in lignified organs for the lignin-related genes comprise the lignin toolbox in mulberry”, but then only genes from Fengchi were analyzed; finally, which genes are considered the lignin toolbox in mulberry then?
- Line 350: so, apparently the authors evaluated the expression of the pre-selected lignin toolbox in Fengchi plants upon zinc stress; however, it is well-known that stress lignification may employ different paralogues than developmental lignification; so, the correct design should be to include all genes in this analysis and evaluate which one respond more drastically to zinc stress
- Line 364-365: can the author explain how exactly tandem duplications were analyzed?
- The discussion of your manuscript is very limited; you should better exploit your findings

---

## Round 0.2 · Minor Revisions

Thank you the manuscript is much improved. Please address the minor concerns raised by the second reviewer.

·

Basic reporting

The manuscript is much improved from the first submitted version. The authors addressed most of my concerns and nicely follow my suggestions (which, by the way, they are not obliged to do so...).
I have minor issues, mostly regading the writing.
Please see below.

Experimental design

No remarks.

Validity of the findings

No remarks.

Additional comments

- Title: including the suggesting to use “mulberry” instead of morus, my suggestion for the title is “Genome-wide analysis of the lignin toolbox of mulberry and its role in the response to zinc stress”
- line 24: no need for “and lignin-related genes” here
- line 29: instead of claiming that “We also indicated the important roles of lignin-related genes in response to stress” I suggest something like “We also observed changes in the expression of some of these lignin biosynthetic genes in response to stress”
- line 39: “efficient conversion”
- line 49: “… and provides hydroxycinnamoyl-CoA esters…”
- line 60-61: “CCR and CAD constitute the primary pathway…”
- line 77-79: my suggestion “The availability of the Morus notabilis genome and an increasing number of transcriptomic data for mulberry allows comprehensive genome-wide analyses of lignin biosynthesis genes in this species”
- line 84: please separate these two sentences (also some suggestions for the second sentence); “M. alba is one of the most widely cultivated mulberry in China. M. alba variety Fengchi is a new variety created by Sericultural Research Institute, Chinese Academy of Agricultural Sciences, expected to spread and grow in extreme environment conditions and used as heavy metal hyperaccumulators and forage.”
- lines 90-93: “We also assessed the potential roles of lignin biosynthetic genes in response to stress caused by the excess of zinc in Fengchi and proposed that the promotion of lignification in lignifying organs, associated with the inhibition of lignin deposition in leaves, is an important response to zinc stress in mulberry”
- line 143: the correct abbreviation is RT-qPCR
- line 146: lignified or lignifying? I guess the second fits better
- line 155: “monolignol”
- line 160: Morus alba (not morus)
- line 173-174: I guess the information on the function of PALs with anthocyanin biosynthesis refers to the Arabidopsis PALs, correct? So, a minor change in the sentence is needed: “… are phylogenetically close to AtPAL1 and AtPAL2, which have been reported…”
- line 175: “MnPAL1 and 5 are preferentially expressed in lignifying organs and tissues…”
- line 213-214: this information on the C4H-C3’H complex should appear after the explanation of each enzymes function, not before; in addition, the authors should make clear what functionality this enzyme-enzyme association allows
- line 223: my suggestion here is “Although C3’H was shown to catalyze the conversion of p-coumaric acid into caffeic acid in vitro, further studies demonstrated that its activity in vitro is the conversion of p-coumaroyl shikimate to caffeoyl shikimate”; (please, also cite Schoch et al., 2003; doi: 10.1074/jbc.M104047200)
- line 228: it is not clear which other MnC3’Hs and FcC3’Hs are mentioned here
- line 229: “C3’H” not “C3H”
- line 233: “C3’H” not “C4H”
- line 238: “to change the carbon flux from H to G and S lignin units”
- line 246: “to distinguish”
- line 249: “sequence similarity”
- line 259: “AtCSE was first characterized as lysoPL2, a member of the monoacylglycerol lipase (MAGL) gene family in Arabidopsis”
- line 289: COMT is not considered an enzyme involved in the biosynthesis of G lignin
- line 310-315: I again insist in the fact that searching for “homologues” of SmF5H in Angiosperms does not make sense here; as I mentioned in my previous report, this genes was shown to have evolved independently in Selaginela; therefore, all the apparent “homologues” identified here are very distantly related and do not represent genes with similar function anyhow
- line 332-333: “and converts various cinnamoyl-CoA esters to their corresponding hydroxycinnamaldehydes…”
- line 368-369: this sentence is confusing; maybe the authors wanted to claim “Hierarchical clustering depicted a similar expression pattern for bona fide lignin biosynthetic genes, which differed from that of genes identified as ‘like’ genes”
- line 368-377: my suggestion to the authors is that they include, in this paragraph, their conclusion on their definition of the lignin toolbox in mulberry; in other words, they should list here which were the genes considered the lignin toolbox in this species; additionally, they should highlight these genes in Fig. S6 (maybe by just including an asterisk in front of the gene name); this would facilitate our understanding
- line 379: the beginning of this sentence is hard to understand; can the authors rephrase, please?
- line 384: “Most of these bona fide clade genes (13/22) were down-regulated in leaves.”
- line 412: “leads to oxidative stress”
- line 414-415: this sentence is confusing; please rephrase it

---

## Round 0.3 · Minor Revisions

Thank you for making the additional changes. There are still a number of issues that are outstanding before the manuscript can be accepted. The section editor has brought up a major point; namely that the data set has not been uploaded to NCBI. The following are their comments:

"It is explained that there were a number of candidates studied, but the reader has no way to access the data. The initial dataset may be available at PRJNA660559, but the assembled transcriptome unigenes are not seen there; where are they? The nomenclature pointed to in a provided website does not have the required information. The ease utility of the data the unigenes should be provided in a third-party location such as Genbank where it can be satisfactorily referenced. If data was sorted into tissue expression levels a call for annotation terms need to be developed. There needs to be some fashion provided in a tabular form (not figure) which outlines unigene, transcript class, ontology, data source reference genome coordinates and such terms for the reader to understand what is actually being presented, and to assist navigation of the data being presented. For having the manuscript declared copy-edited there were a number of inconsistent phrase usage, unique terms, and use of italics. The manuscript reads well in general, but the clarity of the data being provided is still in a confusing state. I would recommend major modification to improve following the facts presented. A few edits are provided below, but not fully inclusive.

Journal manuscripts are often scanned by text-mining software that locates and extracts core data elements, like gene function. Adding standard ontology terms, such as the Gene Ontology (GO, geneontology.org) or others from the OBO foundry (obofoundry.org) can enhance the recognition of your contribution and description. This will also make human curation of literature easier and more accurate. None of this was visible."

In addition, there are further edits that need to be made to manuscript which are listed below.
Please take extra care to review the manuscript again for any corrections that might have been missed.
LINE NO: BEFORE / AFTER / [COMMENTS]
23: lignin toolbox / “lignin toolbox” / [italicized? Not a common term.]
43: / ago. / ago (CITATION). / [When was it discovered?]
47: conversionof/ conversion of
57: general phenylpropanoid/ general phenylpropanoid
75: morus, Moraceae/ Morus, Moraceae/ [genus name are capitalized, family names are not italicized]
77: C. sativa/ Cannabis sativa
78: M edicago/ Medicago
82: branch twigs/ branches twigs
83: produced due to/ produced from
84: considered a new energy plant/ considered a potentially new energy crop
96: 2020) Genome-wide/ 2020). Genome-wide
96: de novo/ de novo
130: experiment/ study
133: MS/ Murashige and Skoog (MS)
134: to provide nutrients/ to provide nutrients
135-138: Sentence is awkward - change so that it makes sense.
146: to screen the candidate/ to screen for candidate
166: sequences/ Sequences
180: Mrousdb/ Mrousdb (http://.....)
180: Clean RNA-seq data/ ? /[What is "clean" data?]
186: Genes showed/ Genes that showed
194: significance/ significant
204: Finally, we obtained/ Finally, We obtained
226: MnPAL1 and 5 preferentially/ MnPAL1 and 5 were preferentially
230-231: to form a gene/ forming a gene
250: 4CLs mainly involved/ 4CLs, which are mainly involved
253: expression preference/ preferential expression
261-262: a kind of liverwort/ the liverwort Plagiochasma appendiculatum
286: in vitro/ in vitro
287: shikimate(Abdulrazzak/ shikimate (Abdulrazzak
343: Arabidopsis/ Arabidopsis
381: pre-eminently/ primarily
440: motif-aware analysis?
455: MnCCR4/ MnCAD4
464: missing reference
465: subtrate; however,/ subtrate, however,
510: Mulberry/ mulberry
544: (Figure 7) Mulberry/ (Figure 7). The mulberry
545: Relative high/ Relatively high
547: natural zinc/ natural zinc

---

## Round 0.4 · Minor Revisions

Thank you for making the requested editorial changes. There are still outstanding issues that have not been addressed.

There are currently no reference-able annotations to the sequences added. This is important due to the different tissue/organ expression and developmental expression of the genes being characterized within the manuscript. Information should be provided in a tabular form (not figure) which outlines unigene, transcript class, ontology, data source reference genome coordinates and such terms for the reader to understand what is actually being presented, and to assist navigation of the data being presented.

---

## Round 0.5 · accepted · Accept

Thank you for adding the requested data set.